

# Reconstructing burnt area during the Holocene: an Iberian case study

Yicheng Shen[1,2,3], Luke Sweeney[1,2], Mengmeng Liu[3], Jose Antonio Lopez Saez[4], Sebastián Pérez-Díaz[5], Reyes Luelmo-Lautenschlaeger[4], Graciela Gil-Romera[6], Dana Hoefer[7], Gonzalo Jiménez-Moreno[8], Heike Schneider[9], I. Colin Prentice[1,3], Sandy P. Harrison[1,2]

[1]Leverhulme Centre for Wildfires, Environment and Society, Imperial College London, South Kensington, London, SW7 2BW, UK

[2]Geography & Environmental Science, Reading University, Whiteknights, Reading, RG6 6AH, UK

[3]Department of Life Sciences, Imperial College London, Silwood Park Campus, Buckhurst Road, Ascot SL5 7PY, UK

[4]Instituto de Historia, Centro de Ciencias Humanas y Sociales, Consejo Superior de Investigaciones Científicas, Madrid, Spain

[5] Department of Geography, Urban and Regional Planning, University of Cantabria, Santander, Spain

[6]Instituto Pirenaico de Ecología-CSIC, Avda. Montañana 1005, 50059, Zaragoza, Spain

[7] Senckenberg Research Station of Quaternary Palaeontology, Am Jakobskirchhof 4, 99423 Weimar, Germany

[8] Departamento de Estratigrafía y Paleontología, Facultad de Ciencias, Universidad de Granada, Avda. Fuente Nueva S/N, 18002 Granada, Spain

[9]Institut für Geographie, Friedrich-Schiller-Universität Jena, Löbdergraben 32, 07743 Jena, Germany

*Correspondence to*: Yicheng Shen (y.shen@reading.ac.uk)

**Abstract.** Charcoal accumulated in lake, bog or other anoxic sediments through time has been used to document the geographical patterns in changes in fire regimes. Such reconstructions are useful to explore the impact of climate and vegetation changes on fire during periods when the human influence was less prevalent than today. However, charcoal records only

provide semi-quantitative estimates of change in biomass burning. Here we derive quantitative estimates of burnt area from vegetation data in two stages. First, we relate the modern charcoal abundance to burnt area using a conversion factor derived from a generalized linear model of burnt area probability based on eight environmental predictors. Then, we establish the relationship between fossil pollen assemblages and burnt area using Tolerance-weighted Weighted Averaging Partial Least-Squares with sampling frequency correction (fxTWA-PLS). We test this approach using the Iberian Peninsula as a case study

because it is a fire-prone region with abundant pollen and charcoal records covering the Holocene. We derive the vegetation-burnt area relationship using the 29 records that have both modern and fossil charcoal and pollen data, and then reconstruct palaeo-burnt area for the 114 records with Holocene pollen records. The pollen data predict charcoal abundances through time relatively well ($R^2 = 0.47$) and the changes in reconstructed burnt area are synchronous with known climate changes through the Holocene. This new method opens up the possibility of reconstructing changes in fire regimes quantitatively from pollen

records, which are far more numerous than charcoal records.



## 1. Introduction

Fire is an important element in many ecosystems and in the Earth system (Bowman et al., 2009; Resco de Dios, 2020). It impacts vegetation dynamics, ecosystem functioning and biodiversity (Harrison et al., 2010; Ward et al., 2012; Keywood et al., 2013). It also affects climate through vegetation changes and the release of trace gases and aerosols. Fire directly impacts

socio-economic assets (e.g. Stephenson et al., 2013; Thomas et al., 2017) and has deleterious effects on human health though releasing smoke and particulates into the atmosphere (e.g. Johnston et al., 2012; Yu et al., 2020). The occurrence of fire is influenced by climate, vegetation properties and human activities. Analyses of the controls on fire based on satellite records can only examine a short time period (ca 20 years). The relationships between fire, natural factors and human influences are still a matter of debate (e.g. Brotons et al., 2013; Bistinas et al., 2014; Knorr et al., 2014; Andela et al., 2017; Forkel et al.,

2019a, 2019b).

Reconstructing changing fire regimes before the Industrial Revolution provides an opportunity to investigate the controls on fire over timescales when human influences on the landscape, including fire regimes, were less pervasive than today. Sedimentary charcoal, preserved in lakes, peatbogs and other anoxic environments, has been widely used as an indicator of

past changes in fire regimes (Marlon et al., 2008, Power et al., 2008; Daniau et al., 2012; Marlon et al., 2016; Vannière et al., 2016; Connor et al., 2019). Evaluations that combine charcoal-inferred palaeofire reconstructions with past hydrological, vegetation, and archaeological data support the idea that there are strong relationships among climate, fire, vegetation and human activities (Carrión et al., 2007; Marlon et al., 2008; Gil-Romera et al., 2010; Turner et al., 2010; Vannière et al., 2011; López-Sáez et al., 2018; Morales-Molino et al., 2018). However, charcoal records only provide a qualitative or semi-

quantitative index of fire activity rather than quantitative estimates of burnt area or biomass loss. Attempts to calibrate the charcoal record to provide quantitative estimates of proximity or area burnt are either site-specific (Duffin et al., 2008; Hennebelle et al., 2020) or rely on modelling (Higuera et al., 2007).

Analyses of present-day fire relationships using satellite-derived data have shown that vegetation properties determining fuel

availability are the strongest determinants of fire occurrence (Bistinas et al., 2014; Forkel et al., 2019a, 2019b; Kuhn-Régnier et al., 2020). This suggests that palaeo-vegetation data could provide a way of reconstructing burnt area in the past, particularly at times when human influences on land cover were less important. Pollen records can be used to reconstruct past climate changes by deriving a statistical relationship between modern pollen abundance and modern climate and applying this relationship to fossil pollen assemblages (Bartlein et al., 2011). Weighted Averaging Partial Least-Squares (WA-PLS)

regression is widely used for this (ter Braak et al., 1993; ter Braak and Juggins, 1993; Salonen et al., 2012). Tolerance-weighted Weighted Averaging Partial Least-Squares with a sampling frequency correction (fxTWA-PLS: Liu et al., 2020) is a modification of WA-PLS, designed to reduce the compression of reconstructions towards the centre of the climatic range



sampled by the training dataset by accounting for the climatic tolerances of individual pollen taxa and the frequency (fx) of the sampled climate variable in the training dataset.


In this study, we present a new method to reconstruct quantitative changes in fire regimes over the Holocene. We relate the relative scale of modern charcoal abundance to absolute burnt area using a conversion factor derived from a generalized linear model (GLM) of burnt area probability. We then apply fxTWA-PLS to derive quantitative relationships between pollen assemblages and the charcoal-derived burnt area, and use this relationship to reconstruct burnt area at sites including those

with no charcoal record. The Iberian Peninsula is the most fire-affected region in southern Europe (Jesus et al., 2019; Molina-Terrén et al., 2019). Although the modern fire regime is partly driven by human activities, the patterns also reflect the strong climate gradients across the region. Although much of the Iberian Peninsula has a typical Mediterranean climate, parts of the region are influenced by proximity to the Atlantic Ocean or the Mediterranean Sea and by the mountainous topography, giving rise to complex weather and climate patterns and large gradients in vegetation diversity (Loidi, 2017). These diverse climate

and vegetation patterns make Iberia a useful test case to explore fire-vegetation relationships. We reconstruct fire regimes across the Iberian Peninsula through the Holocene and discuss the implications of the reconstructed changes.

## 2. Methods

### 2.1. Iberian pollen and charcoal data

Pollen data were obtained from the European Pollen Database (EPD, www.europeanpollendatabase.net) or provided by the

authors. Non-pollen palynomorphs (e.g. fungi, algae), introduced species, and fire-insensitive plants (e.g. obligate aquatics) were removed from the assemblages before analysis. Some pollen taxa are not identified consistently by palynologists or occur at very few sites, so some pollen types were amalgamated to higher taxonomic groups (mostly genera for trees, families for herbaceous taxa) for consistency across the records (Table S1). Charcoal data were obtained from the Global Charcoal Database (Power et al., 2010; Marlon et al., 2016) or provided by the authors. The original age models for both the pollen and

the charcoal records were constructed using different methods and different calibrations of radiometric to calendar ages. We created new age models for all the records using the IntCal20 calibration curve (Reimer et al., 2020) and the BACON Bayesian age-modelling tool in the rbacon package (2.5.0) in CRAN (Blaauw and Christeny, 2011). Charcoal concentration data were converted to charcoal accumulation rate (influx: particles $cm^{-2}$ $yr^{-1}$) before analysis by multiplying concentration with the background sedimentation rate.

### 90 2.2. Development of the Generalised Linear Model

We obtained modern burnt area for the Iberian Peninsula from the fourth version of the Global Fire Emissions Database (GEFD4) (Giglio et al., 2013). The GLM was initially developed using 13 environmental variables covering climate, vegetation and human activities (Table S2). Some environmental data sets were only available at 0.5° × 0.5° resolution, so all data sets



were aggregated to this resolution using bilinear interpolation prior to analysis. Analyses were made for the common period
between the data sets (January 2001 to December 2016) using annual values of all variables. The GLM model was made using
the stats package in R (version R.3.6.0) and used the logit link function and assumed a quasi-binomial distribution (R Core
Team, 2019). We tested combinations of environmental predictors and selected the most parsimonious model with statistically
significant variables and high prediction ability as assessed using pseudo-$R^2$ (McFadden, 1973). The GLM fitted burnt area
was disaggregated from 0.5° × 0.5° to 0.0083° × 0.0083° by bilinear interpolation in order to extract present-day burnt area at
each of the sites with modern charcoal records.

## 2.3. Quantitative reconstructions of burnt area

We derived the relationship between the pollen assemblage and burnt area using 29 records with modern pollen and modern
charcoal (Fig. 1). Rare pollen taxa with ≤ 5 occurrences in the data set were removed because they have been shown to have
little predictive power in WA-PLS climate reconstructions (Turner et al., 2021). The charcoal records included some sites with
only macroscopic and some with only microscopic charcoal. Since this had little impact on the patterns of change through time
(Fig. S3) we used both types, although we used macroscopic charcoal at sites with both macroscopic and microscopic charcoal.
The sampling resolution varies between the individual records. To ensure comparability across records, the charcoal and pollen
data were temporally binned prior to analysis: the modern bin covers the post-industrial period (1850 CE to the present); a
100-year bin width was used for earlier intervals. Pollen counts were summed and converted to percentage of the total count
in each bin. To standardise the values for different charcoal measurement units, we used a max transformation to convert mean
charcoal accumulation rates to a 0-1 range (Eq. (1)).

$$x'_{i,j} = \frac{x_{i,j}}{x_{j,max}} \tag{1}$$

where $x'_{i,j}$ is the transformed value of the i-th sample ($x_i$) in j-th entity. $x_{j,max}$ is the maximum value of all samples in this
entity.

The max transformation resulted in a similar scale of variability between entities in fire-prone areas and areas with little fire.
We therefore applied a conversion factor to re-scale the relative charcoal abundance to absolute burnt area for each of the
records:

$$conversion\ factor_j = \frac{present\text{-}day\ burnt\ area\ fraction_j}{modern\ charcoal\ data_j} \tag{2}$$

where modern charcoal data is the core-top binned charcoal data in the j-th entity and present-day burnt area fraction in j-th
entity was obtained from the GLM.





The palaeo burnt area fraction for i-th sample in j-th entity were then derived by multiplying the conversion factor of j-th entity with the charcoal value for this sample (Eq. (3)):

$palaeo\ burnt\ area\ fraction_{i,j} = conversion\ factor_j \times charcoal\ data_{i,j}$ (3)

We used the fourth root of the palaeo burnt area fraction in the fxTWA-PLS analyses, motivated by consideration of fire spread and the relationship between fire spread and area (see Supplementary Information). The fire-vegetation relationship was determined using the last significant component in fxTWA-PLS, assessed using the p value, to avoid over-fitting.

We applied the fxTWA-PLS derived relationship between pollen abundance and burnt area to the binned pollen data from the 114 pollen records available from the Iberian Peninsula (Fig.1) to reconstruct changes in fire regimes through the Holocene. Some of these 114 entities included pollen taxa that were not present in the data used to derive the vegetation-burnt area relationship; these taxa were therefore removed prior to analysis. We used composite plots and maps of specific times to show the spatial and temporal changes of reconstructed palaeofire regimes through the Holocene. We used loess smoothing with a

window half-width of 300 years to construct the composite plots, with the uncertainty of the reconstruction estimated by bootstrap resampling of the individual reconstructions 1000 times (Efron, 1979; Efron and Tibshirani, 1993). We tested the robustness of our method by comparing the reconstructed burnt area composite with the trends shown by raw charcoal data for those records with fossil charcoal, where the uncertainty is again estimated by bootstrap resampling of the individual charcoal records 1000 times. Maps were created using the reconstructed burnt area for individual sites in the bin covering the

time period of interest.

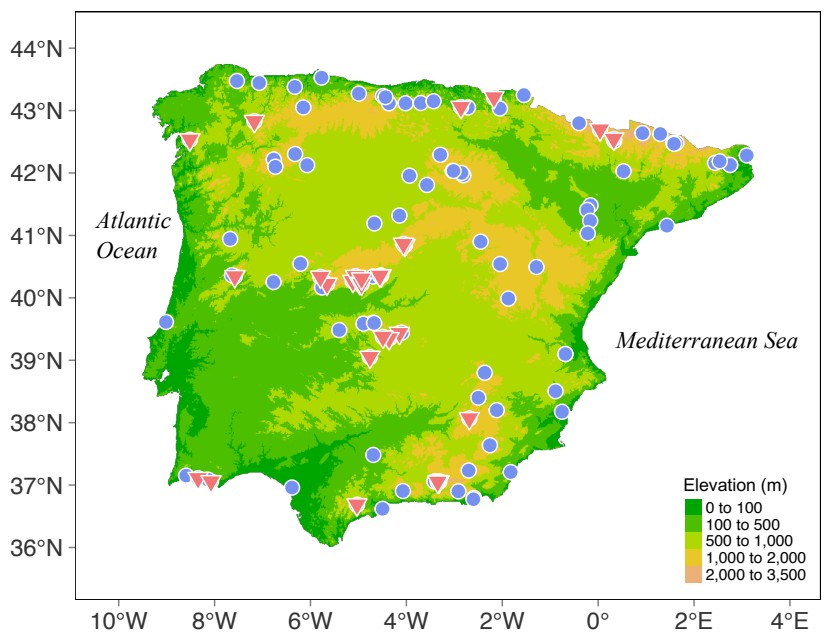



**Figure 1.** Map showing the location of the 29 entities (29 sites) with modern charcoal used to derive the fire-vegetation relationship (red triangles) and the 114 entities (112 sites) used for burnt area reconstructions (blue circles) in the Iberian Peninsula.

**3. Results**

**3.1. The GLM**

Several of the environmental predictors of burnt area were highly correlated to one another (Fig. S1). We tested the impact of including/removing highly and moderately correlated variables before selecting the final GLM (see Supplementary Information). The final model was constructed using eight variables (Table 1, Table S3) and has a pseudo-$R^2 = 0.20$. However,

all of these variables show statistically significant relationships (P < 0.1) with burnt area, and most have p-values < 0.05. Gross primary production (GPP) shows a very strong positive relationship (t = 10.10) with burnt area fraction (Table 1, Fig. S2). Dry days per month (t = 8.46) and non-tree cover (t = 7.34) also show strong positive relationships with burnt area (Table 1, Fig. S2). These relationships make sense given that much of the Iberian Peninsula is relatively arid: increasing GPP and increasing non-tree cover are indices of increased fuel availability in arid, fuel-limited regions and promote increased burnt area. The

number of dry days per month determines fuel dryness, and hence there is a positive relationship between number of dry days and burnt area. Predictions of burnt area from the final model show reasonably good agreement with the observed average burnt area (Fig. 2). The Hotelling's T-Squared Test shows that there is no statistically significant difference between observed and fitted values (p-value = 1). Both the observations and the model show highest burnt area in northern Portugal and moderate burnt area in southern Portugal. Both observed and simulated burnt area are low along the northern coast and in the Pyrenees

where fire is limited by wet conditions, and in the dry interior where fire is limited by fuel availability.

**Table 1.** Generalized linear model of modern burnt area fraction.

| Environmental variable | Regression coefficient (t value) |
|---|---|
| Diurnal temperature range (K) | 1.90. |
| Dry days per month | 8.46*** |
| Wind speed (m/s) | 2.11* |
| Gross primary production (gC m⁻² day⁻¹) | 10.10*** |
| Non-tree cover (%) | 7.34*** |
| Cropland (km²) | -4.04*** |
| Grazing land (km²) | -4.36*** |
| Urban population density (inhabitants km⁻²) | -1.69. |



| | |
|---|---|
| Pseudo-R$^2$ | 0.2031 |

Notes: ·p < 0.1; *p < 0.05; **p < 0.01; ***p < 0.001.

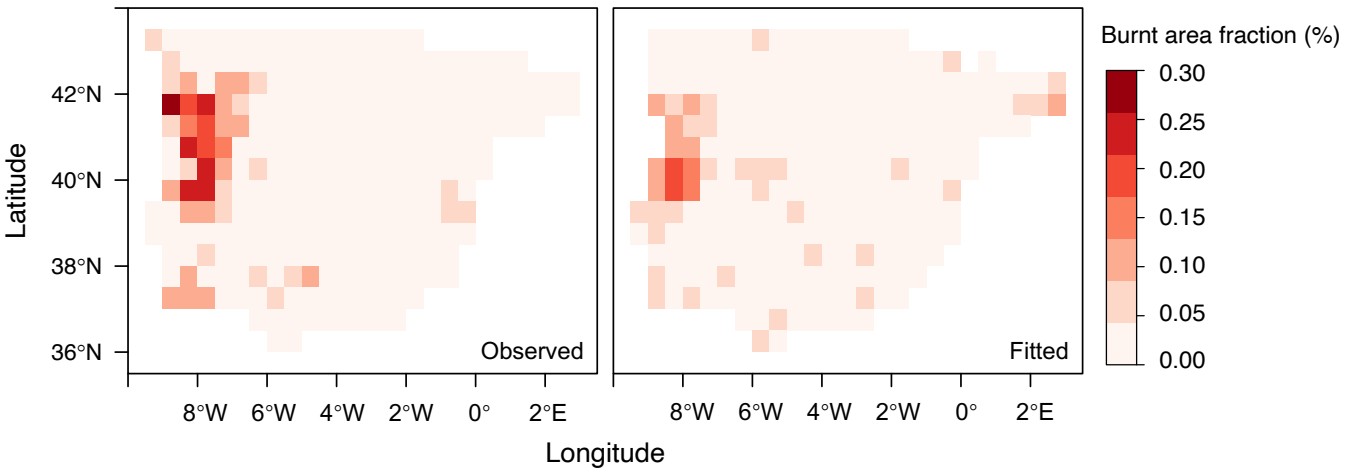

**Figure 2.** Mean (over 16 years) of observed (left) and fitted (right) values of burnt area fraction.

### 3.2 The pollen-burnt area relationship

The fxTWA-PLS derived relationship, based on the last (4$^{th}$) significant component, has good predictive power (R$^2$ = 0.47) (Table 2). A linear regression of the cross-validation results and the burnt area data has a slope of 0.610 (Table 2, Fig 3a) which shows the degree of overall compression towards the centre of the sampled range is relatively low. The degree of local compression, which is assessed by whether the residuals are around zero across the burnt area range in locally estimated scatterplot smoothing, indicates that the low-compression zone where reconstructed values are most reliable is between 0.1$^4$ and 0.2$^4$, in other words, between 0.01% and 16% of the grid cell area (Fig. 3b). Comparison with results using WA-PLS and tolerance-weighted WA-PLS (TWA-PLS) confirms that fxTWA-PLS produces a large reduction in compression in the central part of the burnt-area range and has a higher predictive power (Table S4, Fig. S4, Fig. S5). However, although fxTWA-PLS reduces the compression bias it does not remove it completely: burnt area is overestimated at the low end and underestimated at the high end of burnt area (Fig. 3).

**Table 2.** Leave-out cross-validation fitness of fxTWA-PLS method, showing results for all the components. The last significant number of components are shown in bold. RMSEP is the root mean square error of prediction. ΔRMSEP is the percent change of RMSEP using the current number of components than using one component less. b$_0$, b$_1$, b$_0$.se, b$_1$.se are the intercept, slope, standard error of the intercept, standard error of the slope of the linear regression using the cross-validation result and burnt area data conversed from charcoal abundance.





| Method | ncomp | $R^2$ | RMSEP | ΔRMSEP | p | $b_0$ | $b_1$ | $b_0$.se | $b_1$.se |
|---|---|---|---|---|---|---|---|---|---|
| | 1 | 0.276 | 0.054 | -6.820 | 0.003 | 0.086 | 0.444 | 0.003 | 0.023 |
| | 2 | 0.365 | 0.050 | -7.599 | 0.001 | 0.065 | 0.554 | 0.003 | 0.023 |
| | 3 | 0.435 | 0.047 | -5.311 | 0.009 | 0.057 | 0.628 | 0.003 | 0.022 |
| TWA-PLS with | **4** | **0.467** | **0.045** | **-4.852** | **0.001** | **0.061** | **0.610** | **0.003** | **0.020** |
| *fx* correction | 5 | 0.454 | 0.046 | 1.615 | 0.849 | 0.062 | 0.603 | 0.003 | 0.021 |
| | 6 | 0.468 | 0.045 | -1.735 | 0.005 | 0.062 | 0.606 | 0.003 | 0.020 |
| | 7 | 0.487 | 0.044 | -1.455 | 0.094 | 0.057 | 0.638 | 0.003 | 0.021 |
| | 8 | 0.458 | 0.046 | 4.292 | 0.998 | 0.058 | 0.634 | 0.003 | 0.022 |

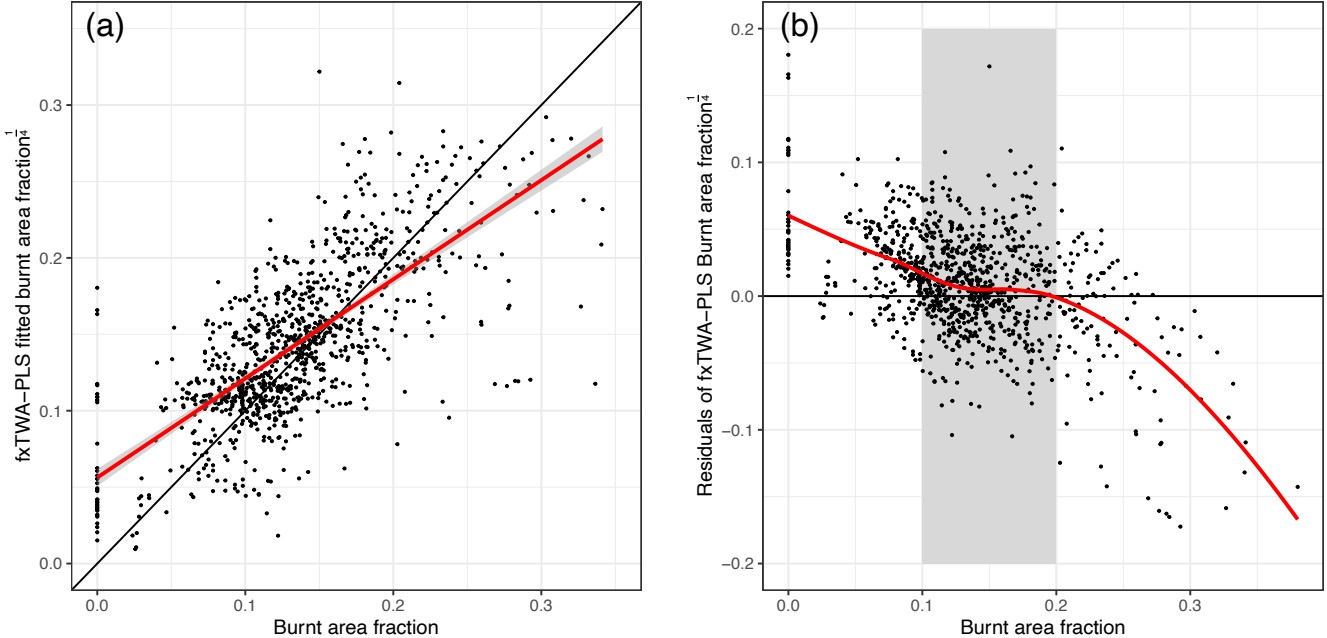


**Figure 3.** The fitted plot and residual plot of TWA-PLS method, with fx correction. Panel (a) is the reconstructed burnt area using the last significant number of components, here is 4. The x-axis is the burnt area fraction derived from charcoal data; the y-axis is the burnt area fraction reconstructed from pollen data using TWA-PLS with fx correction. The 1: 1 line is shown in black; the linear regression line is shown in red, to show the degree of overall compression. Panel (b) shows the residuals of

reconstructed burnt area fraction using the last significant number of components. The x-axis is the burnt area fraction derived from charcoal data; the y-axis is the residual of burnt area reconstruction using TWA-PLS with fx correction. The zero line is





shown in black; the locally estimated scatterplot smoothing is shown in red, to show the degree of local compression; the low compression zone is shown in grey shading.

Charcoal values are not expected to be directly comparable with the reconstructed burnt area but should show comparable temporal trends. A composite plot of reconstructed burnt area for the 22 records which only have fossil charcoal, and therefore cannot be recalibrated, show similar trends to the composite plot derived from the max transformed charcoal (Fig. 4). This suggests there is little distortion of the signal caused by deriving burnt area using the fxTWA-PLS relationship.

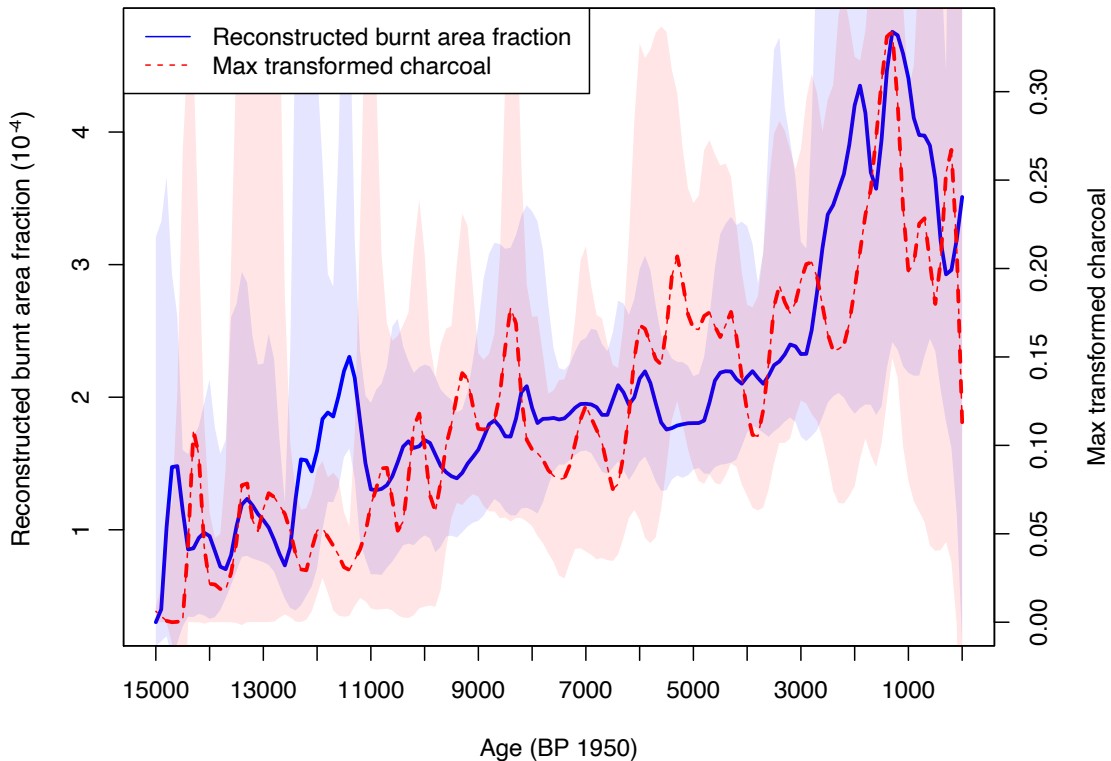

**Figure 4.** Composite plots comparing max transformed charcoal for the 22 records with fossil but no modern charcoal (i.e. records that were not included in the training data set) and the reconstructed burnt area for these records. The reconstructed values, with their uncertainties, are shown in blue; max transformed charcoal, with uncertainties, in red. The smoothing is made with a half-width = 300, and the uncertainty estimated from 1000 bootstrap samples.



### 3.3. Fire history of the Iberian Peninsula through the Holocene

The composite plot based on all 114 pollen records from the Iberian Peninsula (Fig. 5) shows a moderate peak in burnt area around 13 ka followed by a marked increase at 12.5 ka and a subsequent peak in burnt area around 11.5 ka. Although this early part of the record is based on relatively few sites, and so the confidence intervals are large, the pattern corresponds to high fire activity during the Bølling-Allerød (14.6-12.9 ka) warm interval, low fire activity during much of the Younger Dryas (12.9-11.7 ka) cold phase, with an increase in burnt area associated with the rapid warming at the end of the Younger Dryas. Burnt

area is relatively low at the beginning of the Holocene. Although there is a gradual increase in burnt area between 9 ka and 1 ka, the burnt area fraction is lower than present until at least 3 ka. The increase in burnt area is quite marked after 3 ka and peaks around 1 ka. The burnt area fraction at 1 ka is larger than at any time in the record. Burnt area declines after 1 ka, although the modern reconstructed value is still higher than the values obtained for most of the Holocene.

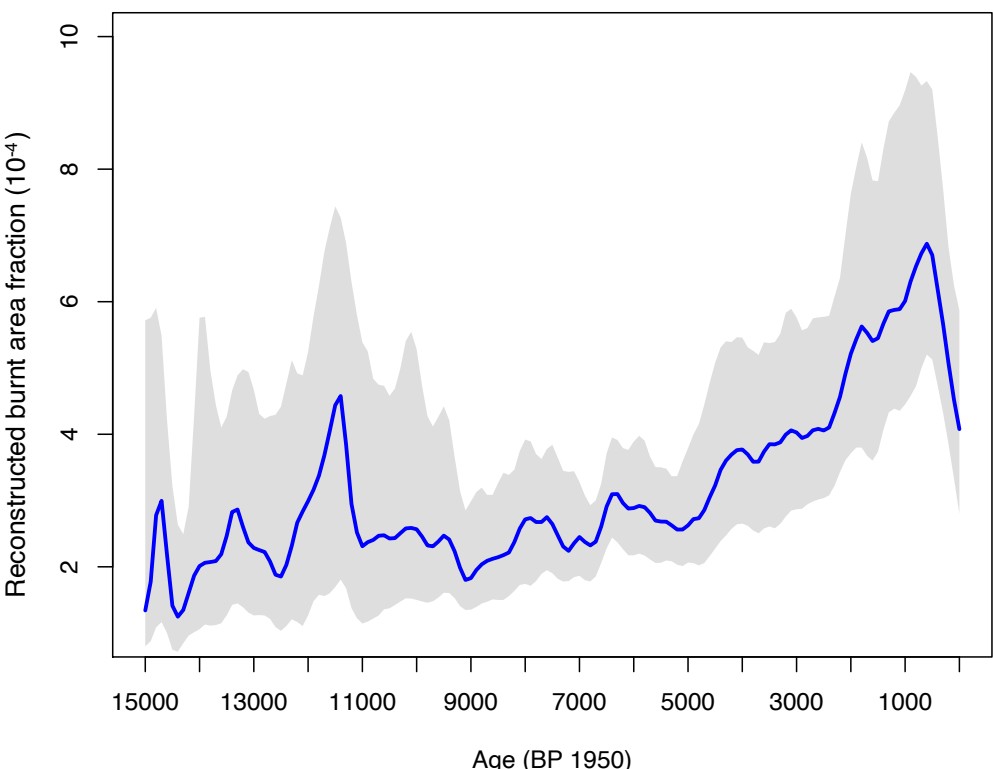

**Figure 5.** Composite curve of reconstructed burnt area using fxTWA-PLS, using the locfit() function with half-width = 300, number of bootstrap samples = 1000. The locally estimated scatterplot smoothing is shown in blue. The upper and lower 95th-percentile confidence intervals are shown in grey.





The spatial coverage of sites (Fig. 6) for the earlier part of the record is sparse, but coverage is good from 7 ka onwards. The
pattern of lower burnt area in eastern than in western Iberia, seen in the modern observations, is generally preserved both in
high and low fire intervals. However, some of the records from northern Iberia (e.g. Saldropo, Puerto de Los Tornos) show
extremely high burnt area which exceeds the scope of the low-compression zone during the last millennium, and particularly
at 0.6 ka. This may reflect the persistent bias at the high end of the fx-TWA-PLS reconstruction range.

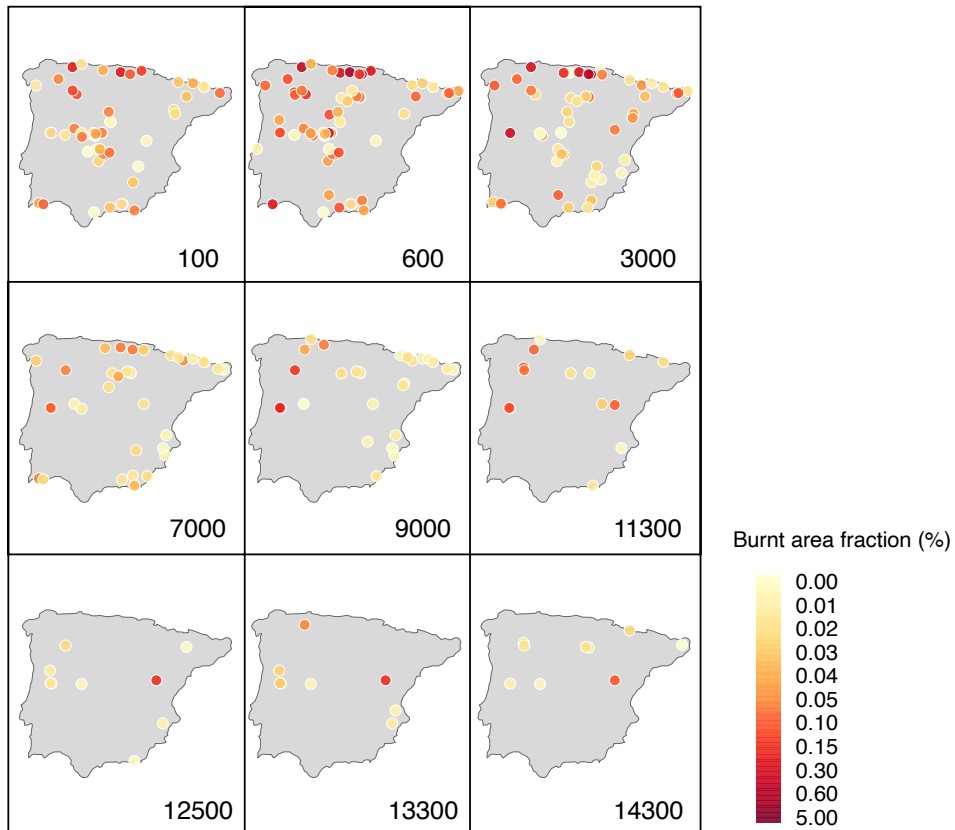

**Figure 6.** Spatial patterns of reconstructed burnt area fraction at key times in the Holocene.

**4. Discussion**

We have shown that it is possible to derive plausible trends in burnt area through time by applying a quantitative relationship
between pollen assemblages and charcoal abundance to palaeo-vegetation records from the Iberian Peninsula. The relationship
between pollen and charcoal is reasonably strong ($R^2 = 0.47$), reflecting the importance of vegetation properties (gross primary
production and non-tree cover) in driving the occurrence of fire as seen in the GLM analysis of the satellite-derived modern
burnt area patterns. The overwhelming importance of vegetation properties in influencing modern fire occurrence is consistent
with results from global analyses (e.g. Moritz et al., 2012; Pausas and Ribeiro, 2013; Bistinas et al., 2014; Forkel et al., 2019b).





Nevertheless, the GLM analysis shows that climate factors, in particular the occurrence of dry intervals, are important controls on modern fire patterns in Iberia. Again, this is consistent with global analyses of the modern drivers of fire occurrence. This

suggests that even stronger reconstructions of changes in burnt area through time could be made by including independent information on climate changes.

We have used fxTWA-PLS (Liu et al., 2020) to make the burnt area reconstructions because this technique reduces the compression bias characteristic of many other reconstruction techniques by accounting for differences in the tolerance of

individual taxa in an assemblage and for the frequency of the reconstructed variable in the training dataset. However, although the bias is apparently reduced, there is still an overestimation at the low end and an underestimation at the high end of the burnt area range. This is reflected by the extremely high burnt area values reconstructed for some sites in northern Iberia in the recent millennium which exceed the upper limit of the low-compression zone. This remaining bias may also be explained by the comparatively small sample size (1015 binned samples) compared with the much larger data set of 6458 samples used by Liu

et al. (2020) to make climate reconstructions for Eurasia. It would be useful to test whether the problem of compression bias in the reconstruction of burnt area could be overcome by expanding the training data set to cover a wider range of vegetation types and fire regimes.

Although palaeo-burnt area reconstructions have only been obtained from a limited number of records, they nevertheless show

interesting patterns over the past ca 15 kyr. The high fire intervals at the beginning of the record, between 14-13 ka and between 12-11 ka, correspond to the Bølling-Allerød (14.6-12.9 ka) warming interval, and to rapid warming at the end of the subsequent Younger Dryas (12.9-11.7 ka) cold phase (Alley et al., 1993). A similar response to these climate events has been seen in charcoal records from eastern North America (Marlon et al., 2009). Burnt area is less than today through the Early and Middle Holocene (10-5 ka), an interval when pollen, speleothem and lake records suggest the Mediterranean region was wetter than

today (Prentice et al., 1996; Magny et al., 2002; Bartlein et al., 2011; Roberts et al., 2011). Reconstructions of fire activity anomalies (FAAs) for the south-eastern part of the Iberia Peninsula also indicate low level fire activity in the mid-Holocene between 7.5 and 6 ka (Gil-Romera et al., 2010). Burnt area continuously increases during the "Medieval Warm Period" (MWP: 1-0.7 ka) and peaks at 0.6 ka, consistent with the warm and dry conditions registered during this period in the Iberian Peninsula (Moreno et al., 2012). During the "Little Ice Age" (LIA: 0.6-0.1 ka), the reconstructed fire indicates a sharp downturn, which

may be associated with subsequent cold and wet climate (Ramos-Román et al., 2016; Abrantes et al., 2017). Thus, the broadscale patterns of trends in reconstructed burnt area are consistent with known Holocene climate changes in this region.

There is a distinct west-east gradient in burnt area across Iberia today, and this gradient of high fire in the west and less fire in the east is also present during other intervals of the Holocene. This pattern is likely related to the regional gradient in fuel

availability and drought (Pausas and Fernández-Muñoz, 2011). However, the west-east gradient in burnt area is less pronounced during the mid-Holocene, consistent with a less pronounced gradient in precipitation and moisture availability



shown by other studies (e.g. González-Sampériz et al., 2017; Liu, 2019). Reconstructed patterns in fire were also more homogenous after 1 ka, and again this is consistent with the fact that temperature and humidity gradients were less pronounced at that time than they are today (Sánchez-López et al., 2016).


Many studies have suggested that human activities have influenced fire regimes during the Holocene (Blanco-González et al., 2018; Connor et al., 2019; Feurdean et al., 2020). Land clearance during the Neolithic agricultural transition has been associated with increases in fire activity in some sites from the Iberian Peninsula (e.g. García-Ruiz et al., 2016; Carracedo et al., 2018). Although initiation of agriculture was not synchronous everywhere, the regional onset of agriculture is registered

around 7.5 ka (Zapata et al., 2004; Fyfe et al., 2019; Harrison et al., 2020) when the burnt area reconstructions do not indicate high fire activity. However, the gradual increase in reconstructed burnt area between 5 and 0.6 ka may be an indication of increasing human activity, since the initial increase is broadly consistent with increased population shown by summed probability distributions (SPDs) of radiocarbon dates (Balsera et al., 2015; Lillios et al., 2016; Harrison et al., 2020). Human activities, such as deforestation and appropriation of land for agriculture, may have been an important driver of fire patterns

from the Bronze Age onwards (Morales-Molino et al., 2013; Morales-Molino and García-Antón, 2014; González-Sampériz et al., 2017), while the competing effects of land abandonment and fire suppression may have contributed to the changes in burnt area in recent times (Turco et al., 2016; Silva et al., 2019). Nevertheless, our GLM analysis indicates that the intensity of human influence, as measured by crop or grazing land area or by population density, consistently has a negative effect on burnt area under modern conditions. It seems likely that human influence on Holocene fire regimes may have been complex, with

agricultural expansion both promoting and suppressing fire occurrence. More detailed comparisons of the reconstructed burnt area and archaeological data are required to test this.

The limited availability of charcoal records, compared for example to pollen records, has meant that the analysis of past fire regimes has tended to focus on large-scale patterns (e.g. Marlon et al., 2008; Power et al., 2008; Daniau et al., 2010; Vannière

et al., 2011). Our new methodology opens up the possibility of reconstructing changes in fire regimes from pollen data and thus of examining finer-scale patterning that might reflect climate or human influences on fire. Spatially explicit reconstructions of burnt area would also be useful to evaluate the simulated response of fire to changing environmental drivers in the past (Thonicke et al., 2005; Brücher et al., 2014; Martin Calvo et al., 2014; Marlon et al., 2016; Kraaij et al., 2020) since comparisons based on qualitative inferences from charcoal are inconclusive (e.g. Brücher et al., 2014).

**5. Conclusion**

We have developed a novel method to reconstruct palaeo-burnt area quantitatively from vegetation records, based on fire-vegetation relationships derived using fxTWA-PLS and the calibration of modern charcoal against GLM modelling of modern burnt area. We have applied this approach to reconstruct changes in burnt area through the Holocene for the Iberian Peninsula.



The good predictive power of the fxTWA-PLS derived fire-vegetation relationship and the plausibility of the palaeofire
reconstructions with respect to known climate changes in the region suggest that this approach could be applied more generally
to provide quantitative reconstructions of past fire regimes.

**Data and code availability.** The pollen and charcoal data from the Iberian Peninsula used in this analysis are available from
http://dx.doi.org/10.17864/1947.294. All other data used are public access. The code used to generate the new age models
(ageR) is available from https://github.com/special-uor/ageR. The code used to generate the fire reconstructions and to create
the composite plots is available from https://github.com/Yicheng-Shen/Palaeofire-reconstruction.

**Supplement.** The supplement related to this article is available online.

**Author contributions.** YS, ICP and SPH designed this study. JALS, SP-D, RL-L, GJM, DH, HS and GG-R contributed pollen
and charcoal data. YS and LS developed the new pollen and charcoal age models. YS carried out the analyses. YS and SPH
wrote the first draft of the manuscript and all authors contributed to the final version.

**Competing Interests.** The authors declare that they have no conflict of interest.

**Acknowledgements.** Yicheng Shen and Sandy P. Harrison acknowledge support from the ERC-funded project GC 2.0 (Global
Change 2.0: Unlocking the past for a clearer future; grant number 694481). I. Colin Prentice acknowledges support from the
ERC under the European Union Horizon 2020 research and innovation programme (grant agreement no: 787203 REALM).
Luke Sweeney acknowledges support from the Leverhulme Centre for Wildfires, Environment and Society. Mengmeng Liu
acknowledges support from Imperial College through the Lee Family Scholarship. José Antonio López-Sáez acknowledges
support from the REDISCO-HAR2017-88035-P (Plan Nacional I+D+I, Spanish Ministry of Economy and Competitiveness)
project. Reyes Luelmo is funded by a FPU grant. Some of the pollen data used in the analyses were extracted from the European
Pollen Database (EPD; http://www.europeanpollendatabase.net/) and the work of the data contributors and the EPD
community is gratefully acknowledged. Some of the charcoal data were extracted from the Global Charcoal Database
(https://www.paleofire.org/index.php), and we gratefully acknowledge contributors to this effort and the curators of the
database. We thank colleagues in the Leverhulme Centre for Wildfires, Environment and Society
(https://centreforwildfires.org/) and from the SPECIAL group at the University of Reading
(https://research.reading.ac.uk/palaeoclimate/) for discussions during the development of this work.

**Financial Support.** This research has been supported by the European Research Council (grant no. GC2.0, 694481), the
European Research Council (grant no. REALM, 787203) and the Leverhulme Centre for Wildfires, Environment and Society,
and the REDISCO-HAR2017-88035-P project.



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
