# Peer review of "Reconstructing burnt area during the Holocene: an Iberian case study"

_Climate of the Past, 2021_

## Community Comment (CC2)

**Response to community comments**
(The reviewer's comments are in regular text and our response in italics)

However I have a major concern regarding the reasoning used in S3 (Justification for the use of the fourth root of the palaeo burnt area fraction used in the fxTWA-PLS analyses)…

*Burnt area data are highly skewed and it is therefore appropriate to transform the data in some way to reduce skewness. We explored various transformation methods and found that the 1/4 power transformation gave good results. However, we agree that the justifiation we gave for this transformation was over-simplified – and it is not necessary to our argument. In response to the reviewer's comment, we have now settled on a Box-Cox transformation with λ = 0.25. We have reanalyzed and updated all the reconstructions results in the latest version of manuscript and explained the process of Box-Cox transformation in the supplementary information:*

*We have modified L125-126 in the manuscript to:*
We applied Box-Cox transformation (Box and Cox, 1964) with λ = 0.25 to the palaeo burnt area fraction in order to reduce skewness prior to the fxTWA-PLS analyses (see Supplementary Information).

*We have modified L170-174 in the manuscript to:*
The degree of local compression, which is assessed by whether the residuals are around zero across the burnt area range in locally estimated scatterplot smoothing, indicates that the low-compression zone where reconstructed values after Box-Cox transformation are most reliable is between –3.5 and –2.5, in other words, between 0.12% and 1.98% of the grid cell area (Fig. 3b).

*In the supplementary information, we have added S4 Box-Cox transformation of palaeo burnt area fraction:*

**S4. Box-Cox transformation of palaeo burnt area fraction**

The standard Box-Cox transformation is:

$$Y_i^{(\lambda)} = \begin{cases} \dfrac{Y_i^{(\lambda)} - 1}{\lambda} & (\lambda \neq 0) \\ \log(Y_i) & (\lambda = 0) \end{cases}$$

The inverse Box-Cox transformation is:

$$Y_i = \begin{cases} \exp\left(\dfrac{\log(1 + \lambda Y_i^{(\lambda)})}{\lambda}\right) & (\lambda \neq 0) \\ \exp\left(Y_i^{(\lambda)}\right) & (\lambda = 0) \end{cases}$$

After deriving paleo burnt area fraction from charcoal by applying conversion factors, we applied the Box-Cox transformation to the palaeo burnt area fraction in order to reduce the skewness of the data. The parameter $\lambda$ was set as 0.25 after trials of a

range of values (Figure S3, Table S7). It shows the highest predictive power ($R^2 =$ 0.472) and a relatively less local compression ($b_1 = 0.549$) comparing with using other values of $\lambda$. Figure S3 shows the change in the distribution of palaeo burnt area fraction before and after Box-Cox transformation. The prediction values of palaeo burnt area fraction from fxTWAPLS were then obtained via the inverse Box-Cox transformation.

[Figure]

**Figure S3.** Distribution of palaeo burnt area fraction before (Panel A) and after (Panel B) Box-Cox transformation.

**Table S7**. Results of fxTWAPLS using different values of $\lambda$.

| Method | ncomp | $R^2$ | RMSEP | $\Delta$RMSEP | p | $b_0$ | $b_1$ | $b_0$.se | $b_1$.se |
|---|---|---|---|---|---|---|---|---|---|
| | 1 | 0.275 | 1.203 | -12.529 | 0.001 | -3.356 | 0.206 | 0.049 | 0.011 |
| | 2 | 0.338 | 1.144 | -4.882 | 0.001 | -3.143 | 0.264 | 0.055 | 0.012 |
| | 3 | 0.421 | 1.051 | -8.128 | 0.001 | -2.558 | 0.409 | 0.071 | 0.015 |
| | **4** | **0.442** | **1.031** | **-1.897** | **0.010** | **-2.337** | **0.461** | **0.076** | **0.016** |
| $\lambda = 0.1$ | 5 | 0.465 | 1.009 | -2.080 | 0.001 | -2.320 | 0.464 | 0.073 | 0.016 |
| | 6 | 0.475 | 1.000 | -0.953 | 0.019 | -2.344 | 0.460 | 0.071 | 0.015 |
| | 7 | 0.477 | 1.009 | 0.904 | 0.700 | -2.511 | 0.414 | 0.064 | 0.014 |
| | 8 | 0.476 | 1.009 | 0.006 | 0.511 | -2.492 | 0.418 | 0.065 | 0.014 |
| | 1 | 0.277 | 0.372 | -13.934 | 0.001 | -2.035 | 0.305 | 0.047 | 0.016 |
| | 2 | 0.363 | 0.350 | -6.013 | 0.001 | -1.668 | 0.442 | 0.056 | 0.018 |
| $\lambda = 0.25$ | 3 | 0.438 | 0.331 | -5.459 | 0.001 | -1.383 | 0.538 | 0.058 | 0.019 |
| | **4** | **0.472** | **0.318** | **-3.815** | **0.002** | **-1.341** | **0.549** | **0.055** | **0.018** |
| | 5 | 0.479 | 0.315 | -1.059 | 0.220 | -1.377 | 0.536 | 0.053 | 0.018 |

| | | | | | | | | |
|---|---|---|---|---|---|---|---|---|
| | 6 | 0.490 | 0.311 | -1.215 | 0.064 | -1.361 | 0.544 | 0.053 | 0.017 |
| | 7 | 0.502 | 0.307 | -1.153 | 0.025 | -1.309 | 0.563 | 0.053 | 0.018 |
| | 8 | 0.510 | 0.306 | -0.419 | 0.311 | -1.252 | 0.580 | 0.054 | 0.018 |
| $\lambda = 0.33$ | 1 | 0.260 | 0.246 | -11.846 | 0.001 | -1.648 | 0.334 | 0.045 | 0.018 |
| | 2 | 0.349 | 0.231 | -6.031 | 0.001 | -1.357 | 0.460 | 0.050 | 0.020 |
| | 3 | 0.429 | 0.216 | -6.631 | 0.001 | -1.168 | 0.534 | 0.049 | 0.019 |
| | **4** | **0.459** | **0.210** | **-2.895** | **0.011** | **-1.105** | **0.557** | **0.048** | **0.019** |
| | 5 | 0.452 | 0.212 | 0.948 | 0.766 | -1.111 | 0.553 | 0.049 | 0.019 |
| | 6 | 0.470 | 0.207 | -2.239 | 0.004 | -1.092 | 0.564 | 0.048 | 0.019 |
| | 7 | 0.480 | 0.206 | -0.523 | 0.197 | -1.045 | 0.584 | 0.049 | 0.019 |
| | 8 | 0.475 | 0.207 | 0.642 | 0.754 | -1.046 | 0.581 | 0.049 | 0.019 |
| $\lambda = 0.5$ | **1** | **0.215** | **0.121** | **-3.625** | **0.054** | **-1.252** | **0.302** | **0.034** | **0.018** |
| | 2 | 0.304 | 0.110 | -9.140 | 0.001 | -1.114 | 0.384 | 0.034 | 0.018 |
| | 3 | 0.370 | 0.106 | -3.367 | 0.043 | -0.933 | 0.481 | 0.037 | 0.020 |
| | 4 | 0.398 | 0.103 | -2.837 | 0.001 | -0.908 | 0.495 | 0.035 | 0.019 |
| | 5 | 0.399 | 0.102 | -0.558 | 0.338 | -0.868 | 0.519 | 0.037 | 0.020 |
| | 6 | 0.417 | 0.100 | -2.099 | 0.003 | -0.832 | 0.541 | 0.037 | 0.020 |
| | 7 | 0.416 | 0.101 | 0.326 | 0.654 | -0.809 | 0.555 | 0.038 | 0.021 |
| | 8 | 0.420 | 0.100 | -0.582 | 0.261 | -0.819 | 0.549 | 0.038 | 0.020 |

Figure 2 indicates the mean annual burned area to reach a maximum of 0.30% in northwestern Iberia. This is 2 orders of magnitude lower than the observed values. Judging from the map in Giglio et al. 2013 it's not produced by GFED4 underestimation, which by the way appears as GEFD4 in the text at least once.

*We have corrected the spelling of GFED4.*

*There are three main differences between the results from Giglio et al. 2013 and the results presented in our manuscript: (1) the time period covered; (2) the spatial resolution; and (3) the definition of "mean area burnt". Here, we used burnt area data covering the period from 2001.01 to 2016.12, whereas Giglio et al. used data from 1996.07 to 2012.08 (their Figure 2). The data in Giglio et al. (2013) are at 0.25° × 0.25° resolution; since some of the environmental data sets we used in our study were only available at 0.5° × 0.5° resolution, we aggregated the burnt area data to this lower resolution using bilinear interpolation. The spatial aggregation has only a minor effect on the estimated burnt area but the choice of time period had a larger effect because our data set includes two years (2008 and 2014: 2014 is not included in Giglio et al. 2013) with a low incidence of fire.*

*However, the major difference between the two maps is caused by differences in the method of calculation of burnt area. We used the mean burnt area fraction of 16 years (192 layers) calculated from the raw monthly data. If this is multiplied by 12 to*

*calculate the annual mean for the interval from 1996-2012, then the overall pattern of burnt area fraction (see Figure below) is very similar, and we obtain a mean burnt area fraction of 0.39%. This value is similar to the estimate of 0.48% land area burnt obtained by Nunes et al. (2019) from forest inventory data. Since our use of averages based on the monthly burnt area is obviously confusing, we will present the results as an annual means calculated following the method of Giglio et al. (2013).*

[Figure]

**Figure extra.** Mean annual area burned, expressed as the fraction of each grid cell that burns each year, derived from the July 1996 to August 2012 monthly GFED4 burned area time series.

*We will replace Figure 2 in the manuscript by this new version:*

[Figure]

**Figure 2.** Mean (over 16 years) of observed (left) and fitted (right) values of burnt area fraction.

*As modern burnt area fraction has changed, Figure 5 and Figure 6 showing reconstructed burnt area fraction have been updated too.*

[Figure]

**Figure 5.** Composite curve of reconstructed burnt area using fxTWA-PLS, using the locfit() function with half-width = 300, number of bootstrap samples = 1000. The locally estimated scatterplot smoothing is shown in blue. The upper and lower 95th-percentile confidence intervals are shown in grey.

[Figure]

**Figure 6.** Spatial patterns of reconstructed burnt area fraction at key times in the Holocene.

[Figure]

Annual burned area of the Iberian Peninsula is currently about 200 kha, or 0.34% of the land mass. However, Fig. 5 points to about 0.04%, so about 10 times less. Again, this does not seem to be an artifact of using GFED4.

*As explained in our response above, this is because we were using the monthly mean values rather than annual mean values. This has now been changed and the values are more consistent with the reviewer's expectations.*

---

## Author Comment (AC1)

**Response to comments by Anonymous Referee #1**
(The reviewer's comments are in regular text and our response in italics)

The calibration of area burnt using modern charcoal is not well explained. There is quite limited information about the modern charcoal samples, which seem to be core tops of unknown time coverage. This information if crucial to assess the validity of the approach. Also, at L99 it says that interpolation was used to extract present-day burnt area at each of the sites with modern charcoal records. However, the comparison of the locations of the modern charcoal samples with the GLM output suggests relatively low spatial coverage of the calibration (e.g. a large area burnt fraction was derived using the GLM in north-central Portugal, but there is only one modern charcoal sample in the region). Another figure showing the calibration of area burnt/modern charcoal needs to be presented, at the moment it is unclear how these two match.

*In this study, modern charcoal bins or "core-tops" cover the post-industrial period (1850 CE to the present) as stated in L108.*

*We realise that the reviewer has misunderstood our approach because we had not explained it clearly enough. In essence, we have derived a relationship between the vegetation assemblage and normalised charcoal using data from multiple sites through time. We then use this relationship to predict fire from vegetation data. However, we need a conversion factor in order to transform the normalized and qualitative charcoal records into a quantitative estimate of burnt area at each site. This conversion factor is derived by relating normalized modern charcoal to the actual observed burnt area via a statistical model. To make this clearer, we have modified the final paragraph of the Introduction (see below). We will also provide a general statement about the approach at the beginning of the Methods section, and include a flow-chart to illustrate the procedure, as follows:*

The central premise of our approach is that fire frequency is one of the factors that influences vegetation assemblages (see Supplementary Information), and therefore that specific aspects of differences in vegetation assemblages – identified by a numerical technique that can isolate the effects of any one controlling factor on taxon composition – can be used to reconstruct fire. The vegetation-fire relationship can be derived by comparing changes in pollen assemblages and charcoal records through time. However, since the charcoal records from different sites consist of different size fractions and the records must be normalised to facilitate comparisons, it is necessary to derive site-specific conversion factors between modern charcoal abundance and present-day burnt area fraction. This calibration is then applied to the charcoal record in order to derive an estimate of the palaeo-burnt area for each pollen sample.

[Figure]

**Figure 1.** Flow chart of the methodology.

The area burnt fraction reconstruction shown in Figure 4 only 'matches' charcoal if we consider a long-term trend perspective. The individual wiggles are often anti-phase, which

raise questions about the validity of the approach. Please reconsider the robustness of this validation.

*We agree that the original figure emphasised the similarity of the long-term trend and that there are some anti-phased relationships on shorter time scales. This could arise because of different numbers of samples in the two data sets, or it could be an artefact of inappropriate choice of span (or half-width) in the loess smoothing. We have explored this by focusing on entities that have both pollen and charcoal records in order to compare the trend of reconstructed burnt area (from the pollen) with the trend of shown by the raw charcoal data. There are 2368 charcoal samples and 2376 reconstructed burnt area samples. If we only consider samples from shared age bins, there are 2104 samples. By using only these 2104 samples we can remove any impact of differences in sampling. We have investigated the impact of using different values of span. It is clear that some of the mismatches in the original plot were due to using an over-large span. Our updated figure, using 0.04 as the span for loess smoothing, shows greater congruence in the placing of peaks (although not in their magnitude). The updated figure is shown below (note that because of the addition of the flowchart, this will now be Figure 5):*

[Figure]

**Figure 5.** Composite plots comparing max-transformed charcoal values and the reconstructed burnt area for these entities for the 51 entities with charcoal. Max-transformed charcoal is shown in blue; burnt area fraction is shown in red. The loess smoothing is made with span = 0.04.

*We have edited L195 to L198 as follows:*

Charcoal values are not expected to be directly comparable with the reconstructed burnt area but should show comparable temporal trends. A composite plot of reconstructed burnt area for the 51 entities that have both pollen records used to reconstruct burnt area and charcoal records, and therefore can be compared, show similar trends to the composite plot derived from the max-transformed charcoal (Fig. 5). This suggests there is little distortion of the signal caused by deriving burnt area using the fxTWA-PLS relationship.

Using pollen data only to reconstruct area burnt can only work in limited conditions. The underlying assumption of the whole methodology is that vegetation (fuel) availability (derived by pollen assemblages) would vary through time only due to fire activity/spread. This assumption does not work in many regions on Earth and I am not quite sure it would

work within some parts of the IP where there are other important controls and where fuel-limitation of fire activity is less dominant. The paper should clarify this limitation and improve its discussion.

*Again there is a misunderstanding of our approach here. We do not assume that the vegetation changes through time only because of fire activity/spread. Our assumption is that changes in the vegetation assemblage can reflect multiple factors,* including *fire regime. We have now quantified this using Canonical Correspondence Analysis to examine how much of the variation in pollen abundances is explained by the environmental factors used in our GLM* (Diurnal temperature range, Dry days per month, Wind speed, Gross primary production, Non-tree cover, Cropland, Grazing land, Urban population density) and how much of this variability can be related to burnt area*. These analyses, which we will include in the Supplementary Information, show that some 19% of the variance in the pollen assemblages is explained by environmental factors other than burnt area but that there is also independent information related to burnt area. We will refer to these analyses in the revised explanation of the methodology that we are adding to the Methods section.*

S1. Canonical Correspondence Analysis (CCA) of the environmental controls on pollen assemblages

The central premise of our approach is that fire is *one* of the factors that modify vegetation assemblages, and therefore that differences in vegetation assemblages can be used to reconstruct fire. We used Canonical Correspondence Analysis (CCA) to investigate how much of the variation in modern pollen assemblages could be explained by burnt area alone, as compared to how much could be explained by burnt area combined with other environmental factors. We used the eight environmental factors considered in our generalised linear model (GLM) for this second analysis, specifically diurnal temperature range, dry days per month, wind speed, gross primary production, non-tree cover, cropland, grazing land, and urban population density. The CCA (Table S1) shows that *ca* 19% of the observed variability in the pollen assemblages is explained by the combination of these environmental variables and burnt area, and that burnt area alone explains *ca* 1% of the variability. Thus, the pollen assemblages contain specific information needed to reconstruct burnt area, even though other environmental influences have larger effects.

**Table S1.** Canonical Correspondence Analysis (CCA) of pollen data

| CCA (a) – burnt area and other environmental variables | | |
|---|---|---|
| | Inertia | Proportion explained |
| Total | 6.189 | 1.0000 |
| Constrained | 1.153 | 0.1862 |
| Unconstrained | 5.036 | 0.8138 |
| CCA (b) – burnt area only | | |
| | Inertia | Proportion explained |
| Total | 6.189 | 1.0000 |
| Constrained | 0.051 | 0.0083 |
| Unconstrained | 6.138 | 0.9917 |

Minor issues:

L30: the assertion that pollen records are more abundant than charcoal records is not valid for many regions on Earth. Maybe this generalisation is true for the Iberian Peninsula, but this has to be clarified.

*There are certainly more pollen records than charcoal records from the Iberian Peninsula (112 sites versus 54 sites). This imbalance is also true globally. There were 736 sites with charcoal records globally in version 3 of the Global Charcoal Database (Marlon et al., 2016) and we currently have a total 1400 sites with charcoal records represented in the Reading*

*Paleofire Database, which is an updated version of the GCD. This compares to 1151 sites with Holocene pollen records from North America (Gajewski et al., 2019), 879 sites with Holocene pollen records from Europe (see e.g. Mauri et al., 2015) alone. Indeed, we do not know of any region where the number of charcoal records available exceeds the number of pollen records. However, since we do not quantify this in the paper, we will modify the last sentence of the abstract to read:*

This new method opens up the possibility of reconstructing changes in fire regimes quantitatively from pollen records, which are often more numerous than charcoal records.

LL32-40: the first paragraph of the introduction reads like a series of loosely connected statements. The rationale for this work needs to be apparent in this paragraph, but at the moment it's quite confusing.

*We agree that the first paragraph only provides a generic statement about why it is important to understand fire regimes. It was designed to lead on to explaining (1) why it is important to look at past fire regimes, and (2) why we exploit vegetation data as a way of doing this. However, we can certainly restructure the introduction to make our rationale clearer. Since many of the comments below also address statements in the introduction, we will first address these and then provide a revised version of this section (see below).*

L42: the Holocene is certainly a period when human agency was 'pervasive' in many regions. This is another generalisation that needs to be better expressed.

*We are not intending to suggest that there was no human influence on the landscape before the industrial revolution, but simply that it was less pervasive than today, and this is true if only because of the much smaller population sizes. We will clarify this by rewriting this statement as follows:*

Reconstructing changing fire regimes during the pre-industrial Holocene (12000 yr B.P. to ca 1850 CE), provides an opportunity to investigate the controls on fire over timescales when human influences on the landscape, including fire regimes, were more localised and less profound than they have become during the industrial era.

L49: remove 'qualitative'

*Much of the literature interpreting charcoal records is indeed qualitative (more fire, less fire). However, we are happy to remove the term and simply say this is a semi-quantitative measure:*

charcoal records only provide a semi-quantitative index of fire activity rather than quantitative estimates of burnt area or biomass loss.

LL54-57: this section does not consider climate into the equation, assuming that fires are fuel-limited. To make it work as a general statement, this should include susceptibility to burn. Alternatively, if this is only referring to the Iberian Peninsula, where fuel availability plays a more important role, this needs to be specified. This is still probably a generalisation, but it works better to introduce the study region.

*Analyses of the drivers of modern fire regimes at a global scale have shown that climate, vegetation and human factors all contribute to determining the incidence of fire. Nevertheless, all of these studies show that vegetation properties, such as primary production and the relative amount of tree versus grass cover, are the most important of these drivers - as we state in this sentence to explain why it should be possible to use*

*palaeo-vegetation data to reconstruct fire histories. Nevertheless, we can expand this text to explain this more clearly as follows:*

Although the occurrence of fire is influenced by multiple factors, analyses of present-day fire relationships globally using satellite-derived data have shown that vegetation properties determining fuel availability are the strongest determinants of fire occurrence (Bistinas et al., 2014; Forkel et al., 2019a, 2019b; Kuhn-Régnier et al., 2020).

LL54-64: this whole paragraph is quite jumpy and confusing (starts with fuel availability, then it goes to pollen assemblages as a method to reconstruct past climates)

*We agree that it is not necessary to introduce the fx-TWAPLS methodology in the Introduction and will remove this.*

LL72-74: This section makes your previous inference about the importance of fuel availability less valid and the whole approach more confusing. I think there needs to be a section introducing the drivers of fires in the IP.

*The modern vegetation patterns strongly reflect climate gradients across the Peninsula, but we agree that we should have made this link to vegetation clearer.*

*Thus, we propose revising the Introduction as follows:*

[revised manuscript text omitted]

L79: this is a general reference to the EPD, but a list of record with references needs to be provided in supporting information

*This information is all included in the data set we provide. However, we will provide a list of records with references in the Supporting Information.*

**Table S2.** Information on the pollen records. Latitude: degrees decimal where +ve is N and –ve is S. Longitude: degrees decimal where +ve is E and –ve is W. Elevation: in metres above sea level. Source: EPD = European Pollen Database (www.europeanpollendatabase.net); PANGAEA = www.pangaea.de/ *(Here, we show the first five rows of the table)*

| Site name | Entity | Source | Latitude | Longitude | Elevation | Reference |
|---|---|---|---|---|---|---|
| Almenara de Adaja | ADAJA | EPD | 41.19 | -4.67 | 784 | (López Merino et al., 2009) |
| Alsa | ALSA | EPD | 43.12 | -4.02 | 560 | (Mariscal, 1993) |
| Alvor Estuary Ribeira do Farelo Ribeira da Torre | Abi 05/07 | author | 37.15 | -8.59 | 1 | (Schneider et al., 2010, 2016) |
| Antas | ANTAS | EPD | 37.21 | -1.82 | 0 | (Cano Villanueva, 1997; Pantaléon-Cano et al., 2003; Yll et al., 1995) |
| Arbarrain Mire | ARBARRAIN | author | 43.21 | -2.17 | 1004 | (Pérez-Díaz et al., 2018) |
| … | … | … | … | … | … | … |

L84: list with references needed for the charcoal records too

*This information is all included in the data set we provide. We will provide a list of records with references in the Supporting Information.*

Table S4. Information on the charcoal records. Latitude: degrees decimal where +ve is N and –ve is S. Longitude: degrees decimal where +ve is E and –ve is W. Elevation: in metres above sea level. *(Here, we show the first five rows of the table)*

| Site name | Entity name | Latitude | Longitude | Elevation | Reference |
|---|---|---|---|---|---|
| Alvor Estuary Ribeira do Farelo Ribeira da Torre | Abi 05/07_100minus | 37.15 | -8.59 | 0.6 | (Schneider et al., 2010, 2016) |
| Alvor Estuary Ribeira do Farelo Ribeira da Torre | Abi 05/07_100plus | 37.15 | -8.59 | 0.6 | (Schneider et al., 2010, 2016) |
| Arbarrain Mire | Arbarrain Mire core | 43.21 | -2.17 | 1004 | (Pérez-Díaz et al., 2018) |
| Armacao de Pera Ribeira de Alcantarilha | ADP 01/06_100minus | 37.11 | -8.34 | 2.4 | (Schneider et al., 2010, 2016) |
| Armacao de Pera Ribeira de Alcantarilha | ADP 01/06_100plus | 37.11 | -8.34 | 2.4 | (Schneider et al., 2010, 2016) |
| … | … | … | … | … | … |

L270: unclear links between paragraphs

*The previous paragraph in the Discussion emphasises the climate controls on vegetation and fire, and the fact that the modern gradients have been present although not constant through the Holocene. However, there is a substantial literature on the potential influence of human activities on fire regimes and we seek to address this here. We can make the transition more apparent by re-writing the first sentence as follows:*

Our analyses show that climate, and climate-induced changes in vegetation, have influenced the fire regimes of the Iberian Peninsula during the Holocene. However, many studies have suggested that human activities could also have been important (Blanco-González et al., 2018; Connor et al., 2019; Feurdean et al., 2020).

L272: this is of course true, but fire is not the only way to achieve land clearance and this approach assumes pollen assemblages are only varying in response to area burnt

*It is true that fire is not the only way to achieve land clearance, although it has been invoked specifically as a method for the Iberian Peninsula e.g. by Connor et al. (2019). This is one reason for looking to see whether there is a relationship between reconstructed fire and the onset of regional agriculture. However, we acknowledge that the onset of agriculture was likely non-synchronous across the Peninsula, and thus the lack of an apparent relationship may hide linkages at a more local scale. However, investigating this possibility requires more detailed local reconstructions of the time sequence for agricultural expansion, and is beyond the scope of the current paper.*

*The reviewer is mistaken in stating that our approach assumes that the pollen assemblages are only varying in response to area burnt. The pollen data are multivariate by nature and this allows us to reconstruct changing fire regimes through changes in the assemblages because some taxa in the assemblage are sensitive to fire. There may however be changes in the assemblages due to e.g. climate and/or human activities, and these changes are independent of the changes in fire regime.*

LL287-289: unclear sentence, confusing how the scarce availability of charcoal records would have led to large-scale patterns (these normally require lots of records)

*Our point here is that the limited availability has meant that analyses of charcoal data have focused either on individual sites or on very broad regions (e.g. continental scale syntheses. However, we agree that our meaning here was not clear and we will re-write this as follows:*

The limited availability of charcoal records has meant that the analysis of past fire regimes has tended to focus on large-scale zonal or continental-scale patterns (e.g. Marlon et al., 2008; Power et al., 2008; Daniau et al., 2010; Vannière 290 et al., 2011). Our new methodology opens up the possibility of reconstructing changes in fire regimes from pollen data and thus of examining finer-scale patterning that might reflect climate or human influences on fire.

---

## Author Comment (AC3)

**Response to comments by Anonymous Referee #2**
(The reviewer's comments are in regular text and our response in italics)

This is an interesting, if challenging, exercise in numerical data transformation. The results are worth reporting, even if the exercise does not appear to have been especially successful.

also final comment: In summary, with modified conclusions, this exercise is worth reporting, but largely because it highlights the difficulties and challenges of using pollen data on their own as a palaeo-fire proxy.

*We disagree that this exercise has been unsuccessful. We have shown that there is independent evidence about fire in the pollen records (please see comments below about the interpretation of the CCA analyses) and that it is possible to reconstruct changes in fire regimes from the pollen data. (Please see additional comment below about the comparisons with charcoal data).*

*Since some of the reviewer's comments suggest a misunderstanding of the approach and how to interpret these results, we will explain this logic more explicitly in the revised manuscript.*

What should be removed is the claim in the conclusion/abstract that "this new method opens up the possibility of reconstructing changes in fire regimes quantitatively from pollen records" in regions where charcoal data are lacking. The pollen-burnt area relationship that they established for the Iberian Peninsula is not transferrable to other geographical regions, even to adjacent regions such as France. For example, wildfire in most of Iberia is fuel-limited so that burning increases at times of wetter climate, when biomass increases. In contrast, in most of France, there is abundant plant biomass so that fires are caused by drought conditions when vegetation becomes more flammable. There are some specific plant taxa that are fire-sensitive or fire-tolerant such as Cistus monspeliensis, but these indicator species are the exception not the norm. In reality, what the authors have done is to use pollen and charcoal data in combination to fill spatial gaps in coverage within Iberia. They have not shown that charcoal can be replaced by pollen as a palaeo-fire proxy in other regions (e.g., Greece) where charcoal data are lacking. The analysis carried out for Iberia could, in theory, be scaled up to cover larger regions, but they are not transferable from one part of the planet to another. Sub-Saharan Africa, for example, is deficient in charcoal records although African savannahs account for almost half of all wildfires globally. This deficiency cannot be resolved by using the pollen-burnt area relationship in, for example, North America, and transferring it to African pollen records.

*The reviewer notes, correctly, that the limitations on fire differ among regions and that it would not make sense to transfer the same relationships from one region to another. However, we are not advocating this. Instead, our analysis shows that there is sufficient information in the vegetation records to derive information about fire. Therefore, the same methodology could be used in other regions, provided that there are entities with both pollen data and modern charcoal data. The absence of charcoal records in some regions, for example the Sahel, means that this approach cannot currently be used to reconstruct palaeo-burnt area there – but there are other regions of the world where it could be applied.*

*To avoid confusion, we will revise the text in both the Abstract and the Conclusions, as follows:*

(abstract) This new method opens up the possibility of reconstructing changes in fire regimes quantitatively from pollen records, after regional calibration of the vegetation-burnt area relationship, in regions where pollen records are more abundant than charcoal records.

(conclusions) The good predictive power of the fxTWA-PLS derived fire-vegetation relationship and the plausibility of the palaeofire reconstructions with respect to known climate changes in the region suggest that this calibration approach could be applied more generally to provide quantitative reconstructions of past fire regimes in other regions where there are limited charcoal data, and pollen data are more abundant.

There are some specific plant taxa that are fire-sensitive or fire-tolerant such as Cistus monspeliensis, but these indicator species are the exception not the norm.

*We disagree with the reviewer. The relative importance of fire-adapted taxa in the vegetation assemblage varies with fire regime (see e.g. Harrison et al., 2021 for an analysis of the abundance of fire-adapted respouting species across Europe in general). The BROT database (Tavsanoglu and Pausas, 2018) provides information about fire-adapted species in the Mediterranean region, including those that are fire-resistant because they have thick bark (e.g.* Quercus suber*), resprouters (e.g.* Juniperus oxycedrus*,* Smilax aspera, Chamaerops humilis, Olea europaea, Arbutus unedo*), taxa that require fire because they are serotinous (e.g.* Pinus halepensis, Pinus pinaster*), and taxa that are stimulated to germinate by smoke or by heat (e.g.* Cistus albidus*,* Cistus monspeliensis*,* Ulex parviflorus*,* Rosmarinus officinalis*). In regions where there is little fire today, the taxa do not display fire adaptations but can be considered sensitive to fire, so there will be a shift in the vegetation assemblage after fire; indeed this is already in some regions where fire frequency has increased recently.*

*The fact that many taxa are fire-adapted makes it possible to derive independent information on fire from the pollen assemblages. We have shown that the variance in pollen assemblages that is explained by fire in Iberia is only 1%, which is substantially less than that attributable to other factors (18%) (climate, vegetation, human activities). This is nonetheless sufficient to be able to exploit the pollen assemblages to reconstruct changes in fire regimes. We expect that in more fire-prone regions, where the abundance of fire-adapted vegetation is greater, the proportion of explained variance would be higher. The fact that it works for Iberia, which as the reviewer points out is not the most fire-prone region of the world, is one reason we suggest our approach could be used effectively elsewhere.*

At a more fundamental level, trying to use pollen data as a fire proxy also means that pollen cannot then be used to test vegetation-wildfire dynamics and relationships, in the way that Connor et al (2019) did for Iberia during the Holocene. Pollen data, on their own, are not able to provide both cause and effect without falling into the trap of circular reasoning.

*On the contrary: the multivariate nature of pollen assemblages means that is entirely possible to infer several quantities simultaneously from the data. Vegetation responds to multiple aspects of the environment, including seasonal climates, fire and other forms of disturbance, and human activities. This has been explored most extensively with respect to climate. Some taxa are particularly sensitive to winter temperatures, for example, while some taxa occur over a wide range of winter temperatures but are sensitive to plant-available moisture. This differential sensitivity to individual climate variables is what allows us to make reconstructions of multiple aspects of the climate from pollen assemblages, and is illustrated in the GAM-based analyses of the climate space occupied by individual European pollen taxa by Wei et al. (2020),* Ecology. *The CCA analyses reported in our paper (Table S1) show that in addition to the climate, vegetation and human influences, the pollen assemblages contain information on fire – thus allowing us to use them to reconstruct fire regimes, without any danger of circularity.*

*We did not explicitly comment on the use of pollen data to test wildfire-vegetation dynamics in our original manuscript (except as a motivation for using Iberia in the Introduction, line 75 in the original manuscript), but nevertheless we do not agree with*

*the reviewer that using the pollen to reconstruct burnt area precludes an analysis of vegetation-wildfire dynamics. Analyses of modern controls on burnt area, including the GLM presented in our manuscript, indicate that gross primary production and the relative abundance of grasses are the most important aspects of the vegetation cover in determining fire occurrence and burnt area (see e.g. Bistinas et al., 2014, Biogeosciences; Forkel et al., 2019, Biogeosciences). Although it has been argued that species composition has an impact on fire regimes in different regions within the same biome (e.g. between North American and Siberian boreal forests), this appears to relate to differences in fire adaptations of individual species rather than being a function of the overall vegetation assemblage. Thus, we argue that it would be useful and interesting to examine wildfire-vegetation dynamics with respect to changing abundance of plant functional types and fire adapted taxa.*

*Since the issue of circularity may be something that concerns other readers, we will add a paragraph in the Discussion about this issue, as follows:*

We have shown that it is possible to derive trends in burnt area through time by applying a quantitative relationship between pollen assemblages and charcoal-derived burnt area to palaeo-vegetation records from the Iberian Peninsula. Our analyses exploit the multivariate nature of vegetation, and hence pollen assemblages. Vegetation patterns, and the distribution of individual species, are controlled by many factors including seasonal temperature and precipitation regimes, disturbance (including wildfires), and human activities. Pollen-based palaeoclimate methods have long exploited the multivariate nature of pollen assemblages to reconstruct different aspects of climate (see e.g. the discussion in Bartlein et al., 2011). The CCA shows that there is sufficient information in the pollen assemblages to assess the independent contribution of fire to vegetation assemblages. The overall relationship between pollen and charcoal-derived burnt area is reasonably strong ($R^2 = 0.47$), reflecting the importance of vegetation properties (gross primary production and non-tree cover) in driving the occurrence of fire – as seen in the GLM analysis of satellite-derived modern burnt area patterns. The overwhelming importance of vegetation properties in influencing modern fire occurrence is consistent with results from global analyses (e.g. Moritz et al., 2012; Pausas and Ribeiro, 2013; Bistinas et al., 2014; Forkel et al., 2019b). Nevertheless, the GLM analysis shows that climate factors, in particular the occurrence of dry intervals, are important controls on modern fire patterns in Iberia. Again, this is consistent with global analyses of the modern drivers of fire occurrence.

How successful was the Iberian test case? Not very successful, as far as I can see. The authors report that pollen data predict charcoal abundances through time "relatively well ($R^2 = 0.47$)".

*The sentence in the abstract led to a misunderstanding here. The reported $R^2$ value is for the relationship between the pollen and burnt area, not the relationship between pollen and charcoal abundance through time. We will revise this sentence in the abstract to:*

The pollen data predict charcoal-derived burnt area relatively well ($R^2 = 0.47$) and the changes in reconstructed burnt area are synchronous with known climate changes through the Holocene.

However, as Figure 4 in the original version of this manuscript shows (Fig. 5 in reply to reviewer comments) this is almost entirely due to a long-term trend during the Holocene towards increased burning. Centennial or millennial scale peaks and troughs in this graph (no longer included in the paper, nor is the helpful flow chart of methodology – why?) are mostly mis-aligned, a point made already by reviewer 1.

*We have not been asked to upload a revised manuscript at this point. However, we propose to include both this Figure (and the flow-chart) in a revised manuscript. As we pointed out in our response to Reviewer 1, the choice of the loess smoothing span has an impact on the shape of the curve. The original span was chosen to emphasise the long-term trends. Furthermore, as we pointed out in our response to Reviewer 1, the number of records included in the two time series was different. The revised figure (now Figure 5) uses only data that are in common between the two data sets, and uses a smaller span for the loess*

*smoothing to more realistically represent shorter-term variations. This revised figure shows better congruence between the placing of peaks.*

In their reply to her/him, the authors also report the results of CCA which shows that burnt area alone explains only 1% of the variability, while other factors explain a much higher share. My guess is that much of this is due to the fact that most of the 29 sites with coupled pollen-charcoal analyses come from two relatively small mountain areas of central Spain (see Figure 1 map in the current manuscript version), so that other regions and biotypes are under-represented in the "training set".

*Indeed, burnt area alone only explains a small proportion of the variability in the pollen assemblages. Nevertheless, this is variability that is not explained by other factors (i.e. it is independent information). Given that vegetation assemblages are controlled by many factors, we do not expect burnt area to explain a high proportion of the variability. Please see our responses above.*

*Although there are several sites from the central mountains of Spain, more than half the records are from outside these regions (see Figure 1 in original manuscript). We agree that the calibration data set is relatively small and it would indeed be worthwhile to test the relationships derived for Iberia over a wider area in order to explore this further.*